# Varying Intensities of Introgression Obscure Incipient Venom-Associated Speciation in the Timber Rattlesnake (*Crotalus horridus*)

**DOI:** 10.3390/toxins13110782

**Published:** 2021-11-05

**Authors:** Mark J. Margres, Kenneth P. Wray, Dragana Sanader, Preston J. McDonald, Lauren M. Trumbull, Austin H. Patton, Darin R. Rokyta

**Affiliations:** 1Department of Integrative Biology, University of South Florida, Tampa, FL 33620, USA; mcdonaldp@usf.edu (P.J.M.); lmtrumbull@usf.edu (L.M.T.); 2Department of Biological Science, Florida State University, Tallahassee, FL 32306, USA; kpwcrotalus@gmail.com (K.P.W.); drags12m@gmail.com (D.S.); drokyta@bio.fsu.edu (D.R.R.); 3Department of Integrative Biology, University of California, Berkeley, CA 94720, USA; austinhpatton@berkeley.edu

**Keywords:** adaptation, venom, genome, ecological speciation

## Abstract

Ecologically divergent selection can lead to the evolution of reproductive isolation through the process of ecological speciation, but the balance of responsible evolutionary forces is often obscured by an inadequate assessment of demographic history and the genetics of traits under selection. Snake venoms have emerged as a system for studying the genetic basis of adaptation because of their genetic tractability and contributions to fitness, and speciation in venomous snakes can be associated with ecological diversification such as dietary shifts and corresponding venom changes. Here, we explored the neurotoxic (type A)–hemotoxic (type B) venom dichotomy and the potential for ecological speciation among Timber Rattlesnake (*Crotalus horridus*) populations. Previous work identified the genetic basis of this phenotypic difference, enabling us to characterize the roles geography, history, ecology, selection, and chance play in determining when and why new species emerge or are absorbed. We identified significant genetic, proteomic, morphological, and ecological/environmental differences at smaller spatial scales, suggestive of incipient ecological speciation between type A and type B *C. horridus*. Range-wide analyses, however, rejected the reciprocal monophyly of venom type, indicative of varying intensities of introgression and a lack of reproductive isolation across the range. Given that we have now established the phenotypic distributions and ecological niche models of type A and B populations, genome-wide data are needed and capable of determining whether type A and type B *C. horridus* represent distinct, reproductively isolated lineages due to incipient ecological speciation or differentiated populations within a single species.

## 1. Introduction

Natural selection can promote local adaptation and determine the geographical distributions of phenotypes within species [1,2]. Ecological speciation is the process by which populations become reproductively isolated due to divergent, ecologically based natural selection [3], and these divergent selective pressures can be due to differences in the abiotic environment [4], assemblages of interacting species [5], or other ecological factors. For example, a reduced immigrant or hybrid fitness mediated through ecological factors could reduce gene flow between populations [6,7], and reproductive isolation would evolve as a result. Evidence in support of ecological speciation is widespread from diverse taxa [3,8,9], but within species, divergence can occur without leading to speciation [10]. Selection can also reduce the extent of reproductive isolation by favoring introgression of mutually beneficial alleles across population and species boundaries (i.e., increased hybrid fitness via heterozygote advantage) [11,12]. Geography, history, ecology, genetics, selection, and chance determine, through unknown relative contributions, whether new species emerge or are absorbed, but the balance of evolutionary forces responsible is generally obscured by the inadequate assessment of demographic history and the genetics of traits under selection [13].

Snake venoms have emerged as a system for studying adaptive diversification because of their genetic tractability [14,15], contributions to fitness [16,17,18], and high evolutionary rates [19,20,21]. Speciation in venomous snakes is often associated with ecological diversification such as dietary shifts and corresponding changes in venom expression [22,23], and gene expression changes have been implicated in ecological speciation in other systems [24]. Rattlesnake venoms are broadly classified into two categories [25]: neurotoxic, which we refer to as type A, and hemorrhagic, which we refer to as type B [26,27]. This dichotomy reflects an inverse relationship between toxicity and tissue-damaging effects [28]. Type A venoms have high toxicity resulting from the presence of a set of homologous potent presynaptic heterodimeric phospholipase A${}_{2}$ (PLA${}_{2}$) neurotoxins (e.g., crotoxin, Mojave toxin, and canebrake toxin) and low levels of tissue-damaging snake-venom metalloproteinases (SVMPs). Type B venoms completely lack heterodimeric PLA${}_{2}$s and express high levels of SVMPs. Dowell et al. [29,30] showed that different sets of PLA${}_{2}$ genes, rather than the differential regulation of shared loci, explained much of the differentiation between A and B venoms (but not all; see [15]).

Most of the >30 rattlesnake species have type B venoms, but type A venoms are known from at least ten species [25]. Nine of these ten species, however, are polymorphic for both venom types. The consistency of the A–B venom dichotomy suggests that intermediate venoms are disfavored by selection (although introgression, albeit rare, has been documented in some cases [31,32,33]), providing a mechanism for intrinsic or extrinsic postzygotic reproductive isolation. Although the selective pressures favoring one venom type over the other are unknown, venoms are costly to produce [34]; selection often favors reduced venom expenditures [18,35], suggesting that type A venoms might be generally advantageous due to their heightened toxicity. The predigestive benefits of type B venoms and SVMPs, however, may allow for the consumption of larger prey or digestion at lower temperatures [25,28], potentially expanding the range, altitude, and diet of type B species and populations. The distributions of A-B venoms in some species have been shown to be correlated with the annual environmental temperature [33], suggesting abiotic as well as biotic (e.g., prey) sources of selection may determine phenotype distributions in this system.

The Timber Rattlesnake (*Crotalus horridus*) occurs in the eastern half of North America from southern Ontario, southward to northern Florida, and westward to Texas and Minnesota [36], exhibiting both A and B venoms [26,27,30,37]. We previously examined venom-gland transcriptomic and venom proteomic divergence between one type A *C. horridus* from Florida (Osceola National Forest, herein referred to as ONF population) and one type B *C. horridus* from Georgia (Jones Center at Ichuaway, herein referred to as JC population) [37]. We found that complete changes in venom composition, rather than point mutations affecting venom gene coding sequences, explained much of the differences between venom types, consistent with the work of Dowell et al. [30]. Dowell et al. [30] used BAC cloning to show that type A *C. horridus* possessed six PLA${}_{2}$ venom genes (including the acidic and basic subunits of canebrake toxin, the defining neurotoxin of type A venoms) whereas type B individuals possessed only one functional PLA${}_{2}$ venom gene, highlighting the role gene loss can play in venom evolution. For SVMPs, however, type B *C. horridus* possessed 13 SVMP paralogs; type A *C. horridus* possessed nearly all of the same loci, indicating that, in this region, the differential gene regulation accounts for the divergent phenotypes, consistent with other type A species [15].

Given that the genetic bases of both venom phenotypes have been characterized in *C. horridus* [30,37], we can now begin to determine the roles geography, history, ecology, selection, and chance play in determining when and why new species emerge or are absorbed. Here, we aimed to (1) establish the phenotypic distributions of A and B venoms, (2) determine whether morphological characters were also associated with feeding such as head shape and fang length [33,38,39] vary by venom type, (3) identify the environmental differences between regions harboring different phenotypes, and (4) determine to what extent the venom dichotomy impacts gene flow. To accomplish these goals, we first sampled individuals from adjacent populations in northern Florida (ONF; type A transcriptome locality) and southern Georgia (JC; type B transcriptome locality) [37] and used proteomic, genetic, and morphological data to extensively characterize microgeographic A–B divergence (i.e., populations were ∼225 km apart). We then sampled *C. horridus* across their entire range to determine the spatial assortment of these phenotypes, explore abiotic characters that may explain these distributions, and test for reciprocal monophyly by phenotype. Overall, we used a combination of data to determine whether type A and type B *C. horridus* may represent distinct, reproductively isolated lineages due to incipient ecological speciation or differentiated populations within a single species. We recognize that mitochondrial-based inferences about species divergence and boundaries can be problematic (see [40]) and consider this study a first-step in exploring the evolutionary relationships between type A and type B *C. horridus*.

## 2. Materials and Methods

### 2.1. Sampling

We collected and/or were provided venom and blood samples from 22 *C. horridus* across Florida (n=7), Georgia (n=7), Texas (n=2), Oklahoma, (n=1), Alabama (n=1), and Ohio (n=4). Snout–vent length (SVL) and total length (TL) were recorded for each individual prior to release. Additional tissue was obtained from 236 preserved specimens that were collected by the authors and/or donated to us for the study (see Acknowledgments section). Samples collected by the authors were conducted under permits LSSC-13-00004, LSSC-09-0399, and LSSC-20-00037 from the Florida Fish and Wildlife Conservation Commission (FWC). All procedures were approved by the Florida State University Institutional Animal Care and Use Committee (IACUC) under protocols #0924 and #1333 and the University of South Florida IACUC under protocol #IS00008815.

### 2.2. Reversed-Phase High-Performance Liquid Chromatography

Reversed-phase high-performance liquid chromatography (RP-HPLC) was performed on a Beckman System Gold HPLC (Beckman Coulter, Fullerton, CA, USA) as previously described [41,42] for the 22 *C. horridus* venom samples described above. In total, 50 μg of venom protein was injected onto a Jupiter C18 column (250 × 4.6 mm; Phenomenex, Torrence, CA, USA) using a solvent system of 0.1% trifluoroacetic acid (TFA) in water (solvent A) and 0.075% TFA in acetonitrile (solvent B). Following five minutes at 5% B, a 1% per minute linear gradient of A and B was run to 25% B; this was followed by a 0.25% per minute gradient from 25% to 65% B at a flow rate of 1 mL per minute. Column effluent was monitored at 220 and 280 nm.

### 2.3. Canebrake Toxin PCR Assay

Genomic DNA was extracted from whole blood samples drawn from the caudal vein or muscle/liver/scale tissue from 258 specimens using the Omega bio-tek E.Z.N.A Tissue DNA Kit according to the manufacturer’s protocol; scale extraction procedures were modified (e.g., increased tissue amounts and addition of 1 M diothiothreitol) as previously described [43]. All extractions were visualized on a 2% agarose gel and run for 30–60 min at 110 V to ensure the presence of high-quality DNA. The acidic and basic subunits of canebrake toxin were amplified from template DNA in 25–50 μL PCR reactions using the acidic subunit-sense (5${}^{\prime }$ GGT ATT TCG TAC TAC AGC TCT TAC GGA 3${}^{\prime }$), acidic subunit-antisense (5${}^{\prime }$ TGA TTC CCC CTG GCA ATT 3${}^{\prime }$), basic subunit-sense (5${}^{\prime }$ AAC GCT ATT CCC TTC TAT GCC TTT TAC 3${}^{\prime }$), and basic subunit-antisense (5${}^{\prime }$ CCT GTC GCA CTC ACA AAT CTG TTC C 3${}^{\prime }$) primers, respectively, under the following thermal cycling protocol: 95 ${}^{\circ }$C for five minutes, 35 cycles of 95 ${}^{\circ }$C for 30 s, 55 ${}^{\circ }$C for 30 s, and 72 ${}^{\circ }$C for five minutes, followed by 72 ${}^{\circ }$C for ten minutes [37,44]. Evidence for amplification was visualized by means of a 0.7% agarose gel and ethidium bromide imaged on a Bio Rad Gel Doc using Quantity One Version 4.1.1 or SYBR Safe DNA gel stain and photographed on Dual LED Blue/White Light Transilluminator.

Because these subunits have been shown to be present in type A individuals but absent in type B individuals [30], we were able to phenotype individuals from only genetic samples. If an assay for an individual amplified both subunits, that individual was considered type A. If an assay for an individual did not lead to amplification of either subunit, that individual was considered type B. No individuals amplified only a single subunit which, based on the genomic location of these genes [29,30], was expected. Because all DNA extractions were visualized on a 2% agarose gel to ensure the presence of high-quality DNA and reduce our false-negative rate, we were confident in all type B classifications; in some instances, amplification of *cytochrome b* (see below) was used to confirm the presence of amplifiable DNA for type B individuals. We do note, however, that this assay could not distinguish between type A and type A+B hybrid venoms, which we discuss below.

### 2.4. Morphological Analysis

We collected data for 5 morphological characters from 33 preserved *C. horridus* specimens (Appendix A) as previously described [38]: (1) SVL, (distance from tip of snout to posterior edge of cloaca), (2) head length (HL, distance from tip of snout to articular-quadrate joint), (3) head width (HW, distance across widest point of head behind the eyes), (4) interfang distance (IF, distance between fangs at maxillae), and (5) fang length (FL, distance from anterior end of maxilla to the tip of fang as folded against roof of mouth). Snout–vent length was measured to the nearest 0.5 cm using a tailor’s tape, and HL, HW, IF, and FL were each measured to the nearest 0.01 mm using 150 mm digital calipers. Venom type was ascribed based on origin from one of the two focal populations (i.e., JC or ONF).

To remove the effect of size prior to statistical analysis, we subtracted each of the four natural-log-transformed head and fang measurements above (2–4) from the natural-log-transformed SVL [45]. Note that for these transformed values, larger transformed values represent smaller raw values and vice versa. We performed Kruskal–Wallis tests on the transformed data for each variable to determine whether type A and type B individuals significantly differed for any character. We then performed a linear discriminant function analysis using the *lda* function and leave-one-out crossvalidation from the MASS package [46] in R to assess group membership placement probabilities based solely on morphology across venom type. All statistical analyses were conducted using R v. 3.6.1 [47].

### 2.5. Niche Modeling

To determine if differences in venom type distributions were correlated with differences in abiotic environmental variables, we followed the approach of Strickland et al. [33] and constructed environmental niche models for each venom type independently using MaxEnt 3.4.1 [48] as implemented in dismo 1.3–3 [49]. We included locality information for 86 type A and 172 type B *C. horridus* and 19 bioclimatic variables as well as elevation at 30 arc second resolution [50]. All predictor variable rasters were cropped to the extent of the presence records and masked with the IUCN species range [51]. To address collinearity, we performed a principal component analysis and removed principal components with eigenvalues <1 [52]. The two highest loading predictors for each of the remaining principal components were included in the model; these variables comprised Minimum Temperature of Coldest Month, Maximum Temperature of Warmest Month, Precipitation of Warmest Quarter, Mean Temperature of Wettest Quarter, Mean Diurnal Range, Temperature Seasonality, Precipitation of Driest Month, Precipitation of Wettest Quarter, and Precipitation of Coldest Quarter. Models were evaluated by AUC with k=4 (i.e., 25% of the data) crossvalidation. To reduce the effects of sampling bias on the model, presence records occurring in the same grid cell were removed. After this reduction, the Type A model used 51 training and 18 testing records, and the Type B model used 85 training and 29 testing records. Variable contribution was assessed by permutation and jackknife.

To test niche equivalency across venom types, we used the niche identity test [53] implemented in ENMTools 1.0.5 [54] to determine whether the distribution of type A and type B individuals was identical in geographic and environmental space. Here, the test computed three measures of similarity between our two environmental niche models, Schoener’s *D*, Warren’s *I*, and Spearman’s rank correlation, and compared these measures across both geographic and environmental space. For each statistic, null distributions were generated using 99 permutation replicates where the occurrence points of type A and type B individuals were randomly reassigned to one of the two models. The test statistic calculated using the original, unpermuted data was then compared with the null distribution to test for significance using a one-tailed test.

### 2.6. mtDNA Sequencing and Phylogenetic Reconstruction

DNA was extracted and quality checked as described above. For 56 *C. horridus*, a 1079 bp fragment of *cytochrome b* was amplified in 25 μL PCR runs using the H16064 and L14910 primers and thermal cycling protocol previously described [55]. PCR products were purified using the QIagen QIAquick PCR Purification Kit. Sequencing was performed on the Applied Biosystems 3730 Genetic Analyzer.

Sequences were aligned and editing using Geneious Prime and adjusted manually. We then constructed two phylogenies using the same approach outlined below: (1) a ONF (n=17) and JC (n=9) phylogeny to match our initial venom and morphological comparisons, and (2) a range-wide phylogeny (n=56). For each, we fit 88 candidate models of nucleotide substitution using jModelTest 2.1 [56] using the CIPRES portal [57]; the best fit model was determined according to the Bayesian Information Criterion. Using BEAST2 [58], we inferred time-calibrated phylogenies. We used a birth–death tree prior as this prior leads to greater accuracy in the estimation of node times when datasets contain a mixture of intra- and inter-specific sampling [59]. For our clock model, we used an uncorrelated lognormal, relaxed molecular clock [60], and estimated, rather than specified, the rate. We specified two fossil calibrations [61] as previously described [38]. Here, we set the age as a mean of 0, SD of 1, and offset these prior distributions using the early boundary of the time period in which the fossils were confirmed. We constrained the outgroup, *C. adamanteus* (sequences from [38]), to be monophyletic, and specified their earliest time to the Most Recent Common Ancestor (MRCA) as 0.781 Ma, using a fossil sample from the Early Pleistocene (Irvingtonian I NALMA: [62]). We constrained the earliest time to MRCA for *C. horridus* to 0.126 Ma based on a fossil sample of this species from the Middle Pleistocene (Irvingtonian II NALMA; [62]). We conducted three independent runs for a length of 50 million generations each, sampling every 5000 generations following an initial burn-in of 10,000 generations. We visually assessed convergence and stationarity for all parameters in Tracer [63]. The three independent runs were subsequently combined, resampling from each every 50,000 generations for a final posterior distribution of 3000 samples. All effective sample sizes for these combined runs exceeded 300. We summarized the posterior distribution of trees using TreeAnnotator to obtain a maximum clade consensus tree using median clade heights for node heights. Trees were visualized in FigTree; clades with posterior probabilities >0.70 were annotated.

## 3. Results and Discussion

### 3.1. Fixed Venom Differences, Deep Mitochondrial Genetic Divergence, and Distinct Morphologies over Small Spatial Scales

To characterize A–B divergence at the microgeographic scale, we used proteomic, genetic, and morphological data to compare the ONF and JC populations, which were only ∼225 km apart. We used RP-HPLC to show that each population was monomorphic for their respective venom phenotype (Figure 1A), with all individuals from JC possessing type B venoms, and all individuals from ONF possessing type A venoms. These fixed differences in venom phenotype were consistent with previous results showing that the two canebrake toxin subunits were present in the genomes of animals from ONF but not in the genomes of animals from JC [37]. Overall, the fixed venom phenotypes between JC and ONF populations suggested little-to-no gene flow, reinforcing the hypothesis that intermediate venoms may be disfavored by selection in this system (but see [31,32,33]) and, therefore, may provide a mechanism for intrinsic or extrinsic postzygotic reproductive isolation.

To determine whether A–B venom differences reflected population structure based on neutral markers at this scale, we constructed a time-calibrated Bayesian phylogeny based on *cytochrome b* sequence for 17 ONF (type A) and 9 JC (type B) individuals (Figure 1B). The venom phenotypes were reciprocally monophyletic with a median divergence time estimate between type A and type B individuals of ∼1.23 Ma (95% CI = 0.249–2.37 Ma). Type A ONF individuals comprised a strongly supported monophyletic clade (posterior probability =0.99) distinct from that of the type B individuals (for which relationships were less well supported). Based on mtDNA, these populations were genetically distinct and putatively reproductively isolated.

To determine whether venom expression differences were integrated with morphological characters also associated with feeding, we measured HL, HW, IF, FL and SVL for 33 preserved *C. horridus* (18 JC and 15 ONF individuals) and performed Kruskal–Wallis rank sum tests on each morphological character by venom type as previously described [38]. Type A individuals had significantly narrower heads (p=0.023) and significantly shorter fangs (*p* < 0.001) than type B individuals (Figure 2A). Although type A snakes also had shorter heads, the difference was not significant (p=0.096); we identified no differences in IF (p=0.745). A linear discriminant function analysis based on these characters alone accurately placed 80.0% of type A snakes and 94.4% of type B snakes (Figure 2B), indicating a strong association between head morphology and venom type, consistent with previous results in rattlesnakes [38]. Coefficients of linear discriminants indicated that fang length (−12.859) was the most important predictor when discriminating among groups (IF =8.059, HW =−4.217, HL =6.278). Although venom plays a primary role in prey incapacitation and predation in rattlesnakes, morphological characters such as gape and fang length may also determine dietary breadth and feeding ecology [33,38,39]. Margres et al. [38] previously showed that fang length positively covaried with myotoxin (i.e., crotamine) expression in *Crotalus adamanteus* and posited that the increased depth of injection for myotoxin-rich venoms may increase predation success. Here, the significant reductions in head size and fang length were inversely correlated with overall toxicity, suggesting that the (1) depth of injection may not matter for type A venoms (or at least may be less important relative to type B venoms) and (2) narrower heads may be a consequence of prey differences and/or a reduction in the volume of venom needed given the increased toxicity of type A venoms. Interestingly, similar work in type A and type B Mojave Rattlesnakes (*C. scutulatus*) did not find significant differences in HW, HL, IF, or FL between type A and type B individuals, although trends did exist for type A individuals to have longer fangs and potentially wider heads [33], opposite of what was found here. The discordance between A–B *C. horridus* and *C. scutulatus* individuals suggests that these morphological differences may be prey specific rather than venom-type specific.

Overall, we identified fixed venom differences, a deep phylogenetic split dating back >1 Ma (although we recognize that mitochondrial gene trees can be misleading when delimiting species (see [40]) and distinct head morphologies across a rather small geographic area. Previous work by ourselves [37] and others [30] showed that many of the major venom differences between these populations, such as the lack of heterodimeric PLA${}_{2}$ neurotoxins in type B venoms, reflected an absence in the genome of the corresponding genes rather than the differential regulation of shared genes. Together, these results suggest that type A ONF and type B JC *C. horridus* may represent independently evolving lineages, perhaps due to incipient ecological speciation. Alternatively, these differences could reflect incipient allopatric speciation as these populations occur on either side of the Suwannee River, a known biogeographic barrier for continental and peninsular Florida organisms [64], including rattlesnakes [13,38]. The Suwannee Straits ran through the Okefenokee Trough from the Gulf of Mexico to the Atlantic Ocean, repeatedly separating peninsular Florida from the continent [65]. Although the Straits likely closed ∼4 Ma, some evidence suggests that the Straits may have opened briefly as recently as ∼1.75 Ma [38,64]. With our current dataset, we could not distinguish between incipient allopatric and ecological speciation at this scale. *Crotalus horridus*, however, has a very large geographic range, and ecological speciation comes with explicit predictions. Isolation by environment (IBE) should be much greater than isolation by distance (IBD) if ecological differences are leading to reproductive isolation [66]. To determine whether phenotypes were spatially distinct or overlapping, we next sampled more broadly across the range to identify the distributions of each phenotype.

### 3.2. Type A Venom Dominates along the Southern Periphery of the Range

We assayed 258 individuals across the range of *C. horridus* and detected 86 type A individuals and 172 type B individuals (Figure 3). Type A venoms were largely present in the southern periphery of the range, and type B venoms dominated the northern and interior parts of the range. Type A venoms may be common along the western range edge as well, but sampling in this region was too sparse to make a determination. Both phenotypes co-occurred in the coastal plain regions of South Carolina, Georgia (see inset in Figure 3), and Orleans Parish, Louisiana, although sampling in the latter was sparse (three type Bs and one type A). We note that our assay could not distinguish between type A and type A+B “hybrid” venoms as both venom types would possess the canebrake toxin subunits. Therefore, some of the individuals classified as type A in Figure 3 could represent A+B venoms, and one type A individual included here from Jekyll Island, Georgia was a *C. horridus* × *C. adamanteus* hybrid [67].

Venom-based analyses showed that both type A and type B phenotypes were similar across the range (e.g., Florida and Texas type A venoms were very similar; Figure 1 and Figure 4), although there appeared to be more variation within type B venoms based on our limited venom-based sampling (Figure 4). Given (1) the much larger geographic distribution of type B venoms and, therefore, greater variation in abiotic and biotic selective pressures and (2) that type B venoms are much more complex than type A venoms (Figure 1 and Figure 4; [15,23]) and have many more axes on which they can vary, this result was expected.

Overall, venom types were spatially sorted across the range, suggesting an ecological divergence between type A and type B individuals. To test whether differences in venom-type distributions were correlated with differences in abiotic environmental variables, we next constructed environmental niche models for each venom type independently.

### 3.3. Type A and B Individuals Occupy Significantly Different Niches

The distributions of type A and type B individuals were significantly different in both geographic ($D_{Geo}=0.29,I_{Geo}=0.58$, Spearman’s rank correlation =−0.03; *p* < 0.01 for all tests) and environmental ($D_{Env}=0.19,I_{Env}=0.44$, Spearman’s rank correlation =0.33; *p* < 0.01 for all tests) space (Figure 5). For the type A model (AUC: training 0.948, test 0.960; Figure 5A), the mean temperature of the wettest quarter (72.2%) and minimum temperature of the coldest month (14.2%) were the most informative environmental variables. Permutation importance (mean temperature of the wettest quarter${}_{perm}=26.1$%; minimum temperature of the coldest month${}_{perm}=60.8$%) and jackknife testing confirmed these results, with mean temperature of the wettest quarter being the most informative variable during jackknife testing. For the type B model (AUC: training 0.826, test 0.810; Figure 5B), precipitation of warmest quarter (47.4%) and temperature seasonality (24.1%) were the most informative environmental variables. Results were again confirmed by permutation importance (precipitation of warmest quarter${}_{perm}=37.3$%; temperature seasonality${}_{perm}=26.4$%) and jackknife testing, with precipitation of warmest quarter being the most informative variable during jackknife testing.

Temperature appeared to be a key factor in determining the distributions of type A and B venoms in *C. horridus*. Type A habitat suitability was positively correlated with mean temperature of the wettest quarter and minimum temperature of the coldest month, the two most informative variables in the model, whereas these variables had relatively low contributions in the type B model (<10%) and were negatively associated with type B habitat suitability. Previous work showed that the distributions of A–B venoms in Mojave Rattlesnakes (*Crotalus scutulatus*) was correlated with minimum temperature of the coldest quarter and mean temperature of wettest quarter [33], consistent with our results here. Given that type A venoms within polymorphic species appear to occur in warmer parts of the range, perhaps the beneficial predigestive benefits of type B venoms and SVMPs are no longer required for effective predation/digestion along the southern periphery [25,28]. In this case, the more toxic type A venoms may be advantageous in the absence of digestive constraints, but when these constraints do apply, type B venoms may be favored, thus limiting the northward expansion of the type A phenotype.

Type A and B venoms, however, could also be locally adapted for different prey species or genetically distinct populations of the same species, and the relationship with temperature may simply correlate with local prey community composition. For example, previous work in Pacific Rattlesnakes (*C. oreganus*) found that biotic information about local prey species explained more variation in venom expression than abiotic environmental data [68]. These hypotheses, however, are not mutually exclusive; here, type B venoms may be adapted for a certain species of prey and pro-digestive effects whereas type A venoms may be adapted for a different species of prey and also be free of digestive constraint. Although diet information for most pit vipers is sparse, we examined 22 published articles, books, and conference proceedings [69,70,71,72,73,74,75,76,77,78,79,80,81,82,83,84,85,86,87,88,89,90] containing dietary information for *C. horridus* across their range to compare prey items in regions where *C. horridus* was monomorphic for type B to regions where *C. horridus* was monomorphic for type A (regions where *C. horridus* was polymorphic in venom type were excluded). We found 130 type B records but only 10 type A records (Appendix A). Mammals (∼76% of type B records, 60% of type A records) and birds (∼17% of type B records, 30% of type A records) comprised the majority of the diet for each venom type. Although the small sample size in type A prevented statistical comparisons, the largest presence/absence difference was found in rodents belonging to the family Cricetidae. Cricetid rodents (e.g., *Peromyscus*, *Microtus*, and *Sigmodon*) were ∼33% of all prey items recorded for type B animals and were notably absent from the type A records. Type B venoms, therefore, may be locally adapted for Cricetid rodents, and perhaps type A venoms are locally adapted to a different prey species and/or abiotic conditions. Toxicity and digestive assays quantifying these differences in natural prey items are needed to test these hypotheses; although interesting and ultimately necessary to characterize the source of ecological divergence (if present), such work was beyond the scope of the present study.

The identified significant differences in niche space, accompanied with differences in multiple, putatively adaptive phenotypes in venom and head morphology, again suggested incipient ecological speciation. Assessing the degree of reproductive isolation is required when investigating putative species boundaries in wide-ranging taxa, especially along contact zones [40,91]. Therefore, we next sequenced *cytochrome b* across the range, with a focus on the contact zone in the southeastern coastal plain, to determine whether venom types retained reciprocal monophyly (as found in the JC and ONF populations) or if reproductive isolation between venom types broke down at larger geographic scales.

### 3.4. A Lack of Reciprocal Monophyly Suggests Gene Flow between Venom Types across the Range

To determine whether the A–B reciprocal monophyly detected for the ONF and JC populations (Figure 1B) was found range wide, we sequenced *cytochrome b* for 30 type A and 26 type B individuals across the range and constructed a Bayesian phylogeny (Figure 6). Although we identified two well-supported monophyletic clades of type A animals (one including individuals from Florida, Georgia, and South Carolina; one including all three Texas type As), neither venom type was monophyletic, and no well-supported monophyletic clades of only type B animals were identified (Figure 6). This pattern suggested that venom types were not reproductively isolated and is consistent with introgression occurring primarily from type A populations into type B populations. However, more extensive sampling of nuclear genetic data is necessary to test this hypothesis.

Given the lack of reciprocal monophyly based on venom type, we next looked at the spatial distribution of the population structure identified in the phylogeny (Figure 6). We identified six clades with varying levels of support: (1) a well-supported type A-only clade containing all 17 ONF individuals, as well as one individual from Georgia and one from South Carolina (pink clade in Figure 6), (2) a well-supported western clade of mostly type B animals (seven type Bs and two type As) along the Mississippi drainage (cyan clade in Figure 6), (3) a well-supported type A-only clade containing three Texas individuals (black clade in Figure 6), (4) a less supported Georgia–South Carolina clade containing all JC individuals and four other individuals (11 type Bs and 2 type As; orange clade in Figure 6), (5) a well-supported South Carolina clade (three type Bs and four type As; gray clade in Figure 6), and (6) a less supported type B only clade in the southeast (yellow clade in Figure 6). Surprisingly, the type A-only pink clade that contained all 17 ONF individuals and 1 individual each from Georgia and South Carolina was more closely related to the western lineages (cyan Mississippi Drainage and black Texas lineages in Figure 6) than the other three lineages identified in the southeast (yellow, orange, and gray lineages in Figure 6); this relationship, however, was not strongly supported (posterior probability =0.74). While this could indicate that the type A phenotype arose in the west and has since spread east, denser sampling across the genome and the range is needed to determine the degree of reproductive isolation between venom types/populations as well as the origin(s) of the type A phenotype in *C. horridus*. Overall, these results revealed substantial geographic population structure within *C. horridus* that was only weakly associated with venom type.

How putatively distinct lineages interact along contact zones and potential species boundaries is essential for determining whether such lineages are evolving independently as distinct species or collectively as geographically differentiated populations [40]. Based on a mitochondrial marker, type A and type B *C. horridus* were not independently evolving lineages and appeared to readily interbreed in coastal Georgia and South Carolina (Figure 6) despite isolation elsewhere (Figure 1). These equivocal results support varying intensities of introgression between type A and type B individuals across the range, and this variance may be the result of variance in ecologically divergent selective pressures. Here, ecologically divergent selection would be strong between the ONF and JC populations, thus producing substantial genetic divergence and putative reproductive isolation, whereas selection would be weaker in the coastal plain of South Carolina and Georgia (i.e., orange and gray clades in Figure 6) where the evidence of introgression was strong. Although the ecologically divergent selection pressures may act on the venom phenotype as predicted in this study, we could not rule out the possibility of other loci in linkage disequilibrium with the PLA${}_{2}$ venom genes being the actual targets of selection, perhaps for a trait not related to predation and feeding ecology. Additionally, neutral processes due to historical biogeographic events (e.g., the formation of the Suwannee Straits as discussed above [65]) could also produce similar patterns. Given that mitochondrial gene trees are often misleading when making inferences about species boundaries [40], nuclear genomic and ecological data across the range with targeted, dense sampling along contact zones is required to better understand the population structure and adequately assess reproductive isolation. Nevertheless, our current mitochondrial data do not support type A and type B *C. horridus* as distinct lineages range-wide.

### 3.5. Conclusions

We used a combination of genetic, proteomic, morphological, and environmental data to test whether type A and type B *C. horridus* may represent distinct, reproductively isolated lineages due to incipient ecological speciation or differentiated populations within a single species. Although we found significant genetic, proteomic, morphological, and environmental differences between type A and type B *C. horridus* at microgeographic scales (i.e., between ONF and JC populations), the lack of reciprocal monophyly among venom types range-wide suggested varying intensities of gene flow and an overall lack of reproductive isolation across the range. Speciation is a continuous process [92], and speciation with gene flow may be common [93], but resolving species relationships among lineages with complex, reticulate evolutionary histories remains challenging [91]. To address this issue, Marshall et al. [40] recently proposed the use of genome-wide nuclear data and intensive sampling along contact zones to determine whether the lineages of interest are evolving mostly independently (i.e., reproductively isolated with narrow zones of F${}_{1}$ hybrids) or as geographically differentiated populations (i.e., intergradation across wider zones with F${}_{2+}$ hybrids). Such an approach here with increased sampling in the coastal plain regions of South Carolina and Georgia as well as along the Gulf Coast is necessary to disentangle the complex evolutionary relationships between type A and type B *C. horridus*.

## Figures and Tables

**Figure 1 toxins-13-00782-f001:**
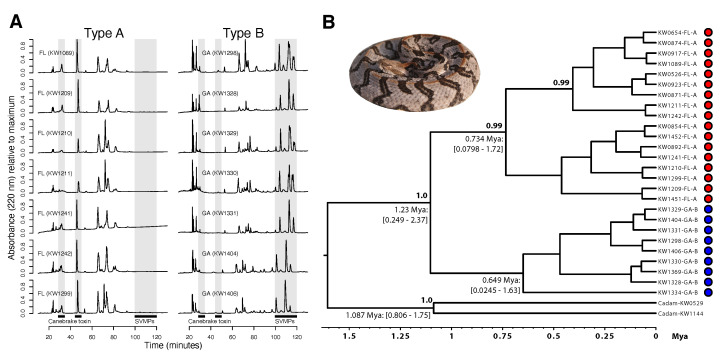
The type B Jones Center (JC) population showed significant venom expression and genetic divergence from the type A Osceola National Forest (ONF) population, suggesting incipient venom-associated speciation. (**A**) Reversed-phase high-performance liquid chromatography (RP-HPLC) profiles for seven adult individuals from each population are shown. The highlighted region from ∼100–120 min includes the snake venom metalloproteinases (SVMPs), and the two phospholipase A${}_{2}$ subunits of canebrake toxin were found prior to 60 min [37]. All individuals from JC were monomorphic for type B venom, and all individuals from ONF were monomorphic for type A venom. (**B**) A time-calibrated BEAST analysis of *cytochrome b* sequences from both populations showed reciprocal monophyly and deep divergence between the populations. Median divergence time estimates (with 95% credible intervals) and clade support values (posterior probabilities) are shown for nodes where the posterior probability >0.70. *Crotalus adamanteus* (Cadam) was used as the outgroup. Type A individuals are represented by red dots, and type B individuals are represented by blue dots.

**Figure 2 toxins-13-00782-f002:**
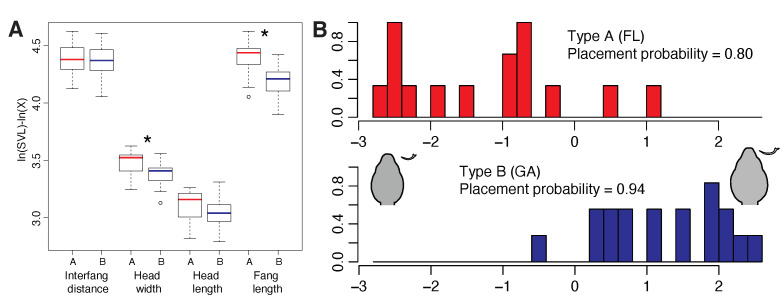
The type B Jones Center (JC) and the type A Osceola National Forest (ONF) populations differed significantly in head morphology. We measured interfang distance, head width, head length, and fang length and adjusted the values on the basis of snout-to-vent length. (**A**) We found significant differences in head width and fang length. The type A population had shorter fangs and narrower heads. Note that due to our transformation of the data, larger values on the *y*-axis represent smaller raw values. (**B**) A linear discriminant function analysis clearly distinguished the two populations on the basis of head morphology. Representative head and fang morphology for each venom type are shown. * p<0.05.

**Figure 3 toxins-13-00782-f003:**
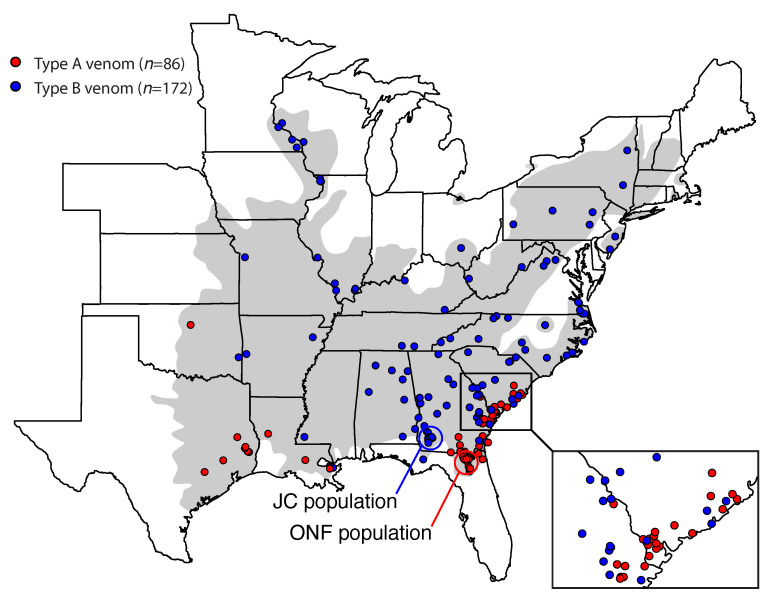
Type A venoms were found largely along the southern periphery of the range. We used a PCR assay to detect the presence of the two subunits of canebrake toxin to establish the presence of type A venom. The grey shading indicates the historical range of *C. horridus*. Our two focal populations (JC and ONF) are indicated. Some points were jittered in an attempt to show all individuals from a single locality. The inset map in the bottom right corner shows a clear contact zone between type A and type B individuals in the coastal plain region of South Carolina and Georgia.

**Figure 4 toxins-13-00782-f004:**
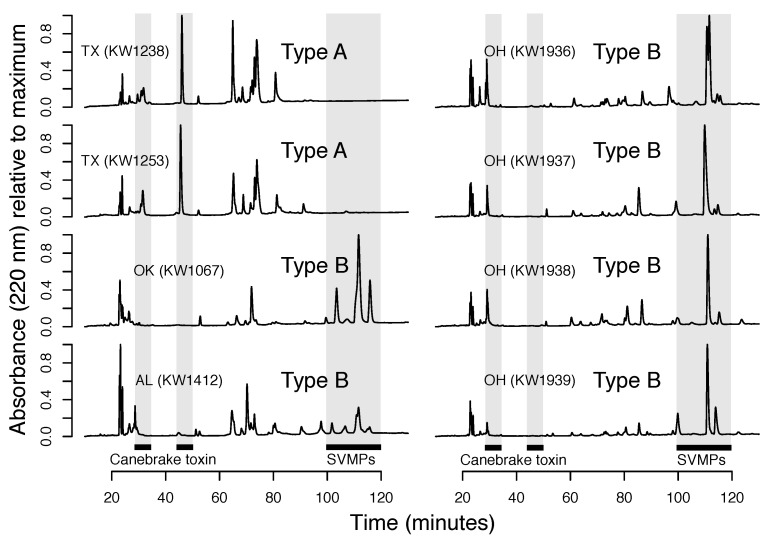
Representative reversed-phase high-performance liquid chromatography (RP-HPLC) profiles of venoms from additional populations showed that the A/B dichotomy was consistent across the range. Venoms either had peaks after 100 min and were therefore type B or peaks before 60 min that corresponded to the two canebrake toxin subunits.

**Figure 5 toxins-13-00782-f005:**
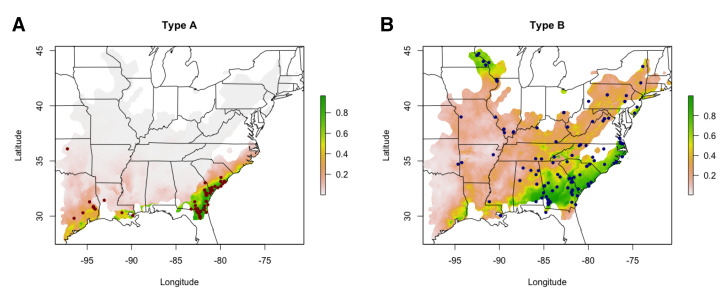
Environmental niche models for Type A and Type B individuals revealed significant differences in geographic and environmental distributions. Habitat suitability predictions produced by MaxEnt environmental niche models using 69 Type A (**A**) and 114 Type B (**B**) presences, indicated by points, are shown. The heatmap corresponds to habitat suitability scores and can be interpreted as a percentage.

**Figure 6 toxins-13-00782-f006:**
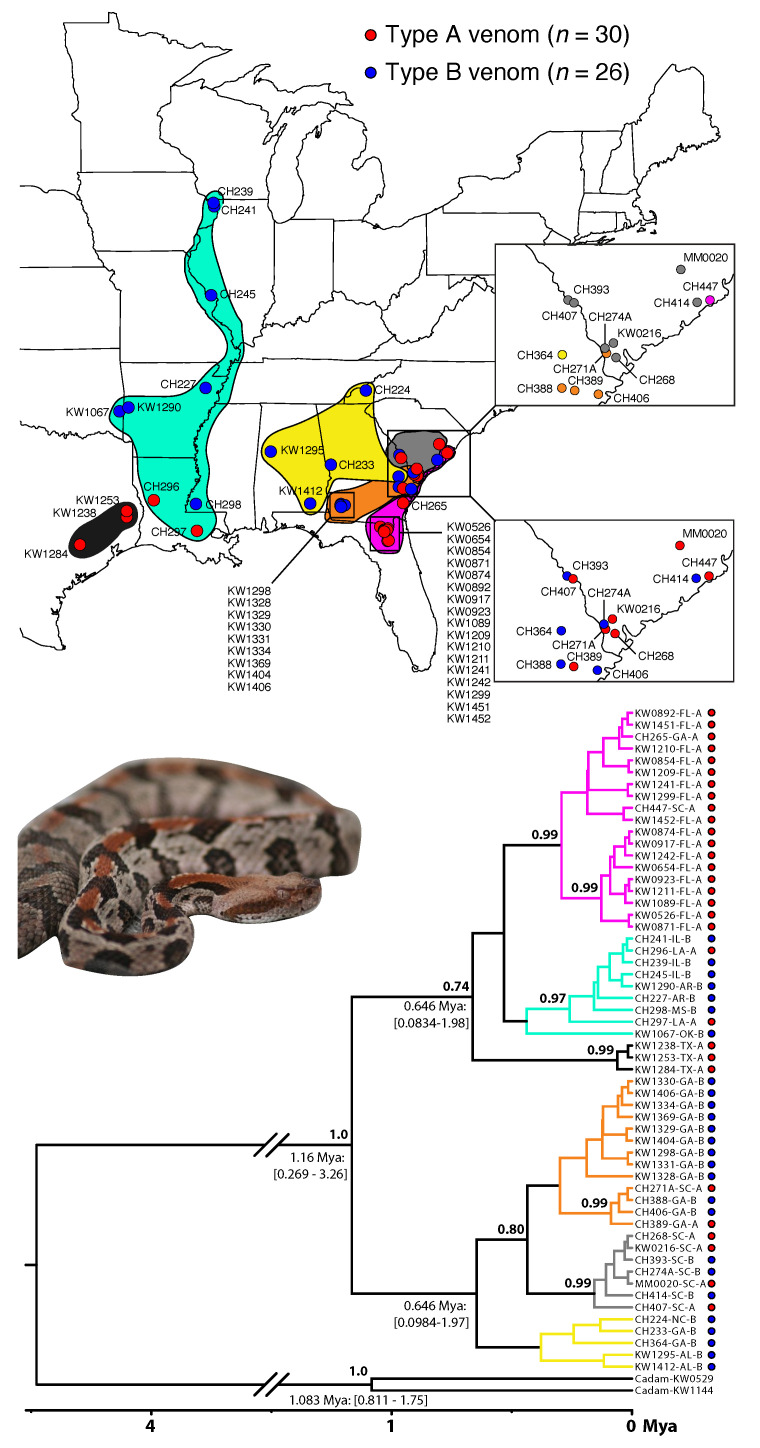
Venom types were not reciprocally monophyletic, suggesting gene flow between venom types in some geographic regions. We sequenced *cytochrome b* from 30 type A and 26 type B individuals and constructed a time-calibrated BEAST phylogeny. We identified two well-supported monophyletic clades of only type A animals, but we found no well-supported pure type B clades. The dot to the right of each tip corresponds to venom type (red type A, blue type B), and branch color corresponds to clade (distribution colored on map above). Top map inset shows clade color; bottom inset shows venom type. Median divergence time estimates (with 95% credible intervals) are shown at key nodes. Posterior probabilities >0.70 are shown. *Crotalus adamanteus* (Cadam) was used as the outgroup.

## Data Availability

All *cytochrome b* sequences used in our phylogenetic analyses were deposited on NCBI under accessions MZ394520–MZ394575.

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
