# Peer review of "Varying Intensities of Introgression Obscure Incipient Venom-Associated Speciation in the Timber Rattlesnake (Crotalus horridus)"

_toxins, 2021, doi:10.3390/toxins13110782_

Round 1
Reviewer 1 Report
The authors have made a valuable contribution to our understanding of the evolutionary processes driving venom divergence.
They have detected 2 venom phenotypes which are largely allopatric (one southern neurotoxic, one northern/interior haemotoxic/tissue destroying). They have detected gene flow at contact zones (South Carolina/Georgia), but that the venom phenotypes remain distinct in these circumstances, suggesting intermediate venom phenotypes are not favoured. It may be an example of geographical variation in venom which became fixed due to geographical factors creating a barrier to reproduction (Suwannee River) that infers a fitness benefit in warm climates in the absence of digestive constraints (although this is only hypothesised at this point, but seems plausible). Geographical variation occurs in snake venoms all the time, but in this instance it may have beneficial enough to contribute to incipient speciation?
The authors did not address the possibility of the venom differences possibly being the result of linkage disequilibrium due to toxin genes being closely associated with another yet unknown but favourable non-venom trait. That is pure speculation on my part (and may not be the case), but the authors may want to consider adding a sentence or two addressing that.
Author Response
We agree with the reviewer that this alternative hypothesis should be presented, and we have included a brief discussion on lines 454-458.

Reviewer 2 Report
In this manuscript the authors investigate evidence for ecological speciation among Timber Rattlesnakes with distinct venom phenotypes (Type A and B). At a fine scale, they identify genetic, morphological, and ecological differences between A and B animals, but analyses of range-wide sampling do not support strong reproductive isolation between animals with different venom types. Overall, I think this manuscript is well written and will be of interest to the readership of Toxins.
Is any information available for the primary prey and diet of Type A vs. Type B animals? And is type B venom more advantageous for securing particular prey types compared to type A, and vice versa? The authors look at associations between venom type and various environmental factors (Section 3.3), and find that temperature is correlated with venom type with B generally being associated with colder environments and A with warmer. They suggest that this may be related to pre-digestive benefits of type B venoms. These interpretations feel as if they are based on the assumption that prey type is consistent throughout the range. However, I would imagine that the type of prey available to and thus potentially the diet of northern/B and southern/A populations would vary substantially, and perhaps itself even be correlated with temperature. Perhaps such diet information is not available, but it seems like the potential for variation in prey/diet to impact predominant venom type should be mentioned and discussed.
I feel that all analyses and interpretations related to gene flow would be substantially improved by nuclear data, as mitochondrial inferences likely only tell part of the story. However, the authors do a reasonable job of highlighting this limitation of the current study, and highlight the value of future integration of genome-wide nuclear data to study this system. I hope that the authors are indeed planning to generate such data for a follow-up study!
Author Response
We agree with the reviewer that dietary differences are likely important in this system and previously highlighted this in the Introduction. To address the reviewer’s comment, we did an extensive literature search and found 22 studies with dietary information for C. horridus and locality information as locality was necessary to compare A and B diets. We have now added a paragraph on lines 374-398 discussing putative differences between regions with only type A snakes (n=10) and regions with only type B snakes (n=130).
We agree with the reviewer that nuclear data is needed, and we have tried to make that abundantly clear in the manuscript. We do hope to use our extensive sampling in this system to perform a genome-wide study in the future.
